# The Role of the Gut Microbiome in Colorectal Cancer Development and Therapy Response

**DOI:** 10.3390/cancers12061406

**Published:** 2020-05-29

**Authors:** Lidia Sánchez-Alcoholado, Bruno Ramos-Molina, Ana Otero, Aurora Laborda-Illanes, Rafael Ordóñez, José Antonio Medina, Jaime Gómez-Millán, María Isabel Queipo-Ortuño

**Affiliations:** 1Unidad de Gestión Clínica Intercentros de Oncología Médica, Hospitales Universitarios Regional y Virgen de la Victoria. Instituto de Investigación Biomédica de Málaga (IBIMA)-CIMES-UMA, 29010 Málaga, Spain; l.s.alcoholado2@gmail.com (L.S.-A.); auroralabordaillanes@gmail.com (A.L.-I.); maribelqo@gmail.com (M.I.Q.-O.); 2Departamento de Cirugía Digestiva, Endocrina y Transplante de Órganos Abdominales, Instituto Murciano de Investigación Biosanitária (IMIB-Arrixaca), 30120 Murcia, Spain; brunoramosmolina@gmail.com; 3Unidad de Gestión Clínica de Oncología Radioterápica, Hospital Universitario Virgen de la Victoria. Instituto de Investigación Biomédica de Málaga (IBIMA), 29010 Málaga, Spain; ana.otero.rom@gmail.com (A.O.); rafaelordm@gmail.com (R.O.); jmedinacarmona@gmail.com (J.A.M.)

**Keywords:** colorectal cancer, gut microbiota, dysbiosis, inflammation, short-chain fatty acids, polyamines, dietary fiber, polyunsaturated fatty acids, polyphenols, probiotics

## Abstract

Colorectal cancer (CRC) is the third most common cancer worldwide and the leading cause of cancer-related deaths. Recently, several studies have demonstrated that gut microbiota can alter CRC susceptibility and progression by modulating mechanisms such as inflammation and DNA damage, and by producing metabolites involved in tumor progression or suppression. Dysbiosis of gut microbiota has been observed in patients with CRC, with a decrease in commensal bacterial species (butyrate-producing bacteria) and an enrichment of detrimental bacterial populations (pro-inflammatory opportunistic pathogens). CRC is characterized by altered production of bacterial metabolites directly involved in cancer metabolism including short-chain fatty acids and polyamines. Emerging evidence suggests that diet has an important impact on the risk of CRC development. The intake of high-fiber diets and the supplementation of diet with polyunsaturated fatty acids, polyphenols and probiotics, which are known to regulate gut microbiota, could be not only a potential mechanism for the reduction of CRC risk in a primary prevention setting, but may also be important to enhance the response to cancer therapy when used as adjuvant to conventional treatment for CRC. Therefore, a personalized modulation of the pattern of gut microbiome by diet may be a promising approach to prevent the development and progression of CRC and to improve the efficacy of antitumoral therapy.

## 1. Introduction

Microbiota is composed of different bacterial populations with a mutualistic relationship that reside in the epithelial barriers of different organs in the host. Microbiota is a metabolically active ecosystem that interacts with epithelial and stromal cells, with a critical role in human health. Microbiota carries out different functions such as the production of diverse important metabolites, the prevention of infestation by pathogens, and the control of the overgrowth of some bacterial groups to prevent the modulation of the local environment by toxic bacteria [1]. In addition, microbiota is essential for the activation of the host immune system [2].

The quantity and diversity of microbial species in the gut increase longitudinally from the stomach to the colon, being the colonic microbiota the most dense and metabolically active community [3]. Although the composition of microbiota is influenced by genetics [4] and may be considered relatively stable within healthy adults over time [5], there is a large variation in the microbiota composition among individuals; this variation is conditioned by different external environmental factors such as diet, chemical exposure, and antibiotic/medication consumption. Dietary changes have been shown to have significant effects in gut microbiota, and the switching from a high-fat/low-fiber diet to a low-fat/high-fiber diet may cause important changes in the gut microbiota within 24 hours [6].

In the last decade numerous works have established a clear relationship between alterations in the gut microbiota composition and diverse human pathologies. In particular, obesity and associated metabolic disorders (e.g., type 2 diabetes and non-alcoholic fatty liver), autoimmune diseases (e.g., type 1 diabetes and inflammatory bowel disease), and several types of cancer are characterized by changes in the microbiome and gut dysbiosis [7]. 

The gut microbiota produces a diverse metabolite repertoire that may harm or benefit the host. Alterations in the intestinal bacteria balance could lead to changes in the levels of gut microbial metabolites such as short-chain fatty acids (SCFAs), polyphenols, vitamins, tryptophan catabolites and polyamines [8], which could be related to the pathogenesis of the human diseases described above. In particular, abnormal levels of SCFAs and molecules related to the metabolism of amino acid like polyamines have been involved in cancer progression and metastasis in different types of tumor [9]. 

In this review we discuss the potential role of gut microbiota in the carcinogenesis of colorectal cancer (CRC), the possible role of bacterial metabolites in CRC development and progression, and the influence that certain dietary mediators exert over the intestinal microbiota and CRC risk. 

## 2. Gut Microbiota and CRC

### 2.1. Gut Microbiota Composition in CRC

CRC is the third most common cancer worldwide; nevertheless its exact aetiology is still unknown [10]. Most of the CRC cases are sporadic (nearly 90%), and some genetic and environmental factors have been identified as potential risk factors. Lifestyle factors that increase the risk of CRC in developing countries include physical inactivity, smoking, unhealthy dietary habits (e.g., diets rich in processed and red meat, high fat diets, low intake of fibre), alcohol consumption, and obesity [11,12,13]. Importantly, all these enviromental factors are able to produce changes in the gut microbiota composition [14].

Emerging evidence suggests that in animal models gut microbiota may contribute to CRC development through the production of microbial metabolites that interact with the host-immune system and induce the release of genotoxic virulence factors [1,2,15,16]. Recent works have reported that patients with CRC display a lower bacterial diversity and richness in fecal samples and intestinal mucosa compared to healthy individuals [17,18]. In addition, CRC patients show significant alterations in specific bacterial groups with a potential impact on mucosal immune response with respect to healthy controls [18]. In particular, CRC patients exhibit a significant increase in *Bacteroides fragilis*, *Fusobacterium nucleatum*, *Enterococcaceae* or *Campylobacter, Peptostreptococus, Enterococus faecalis, Escherichia coli, Shigella* and *Streptococcus gallolyticus*, and a decrease in *Faecalibacterium*, *Blautia, Clostridium, Bifidobacterium* and *Roseburia* [19]. These changes might produce enrichment in pro-inflammatory opportunistic pathogens and a decrease in butyrate-producing bacteria, which may lead to an imbalance in intestinal homeostasis (dysbiosis) that could ultimately lead to tumor formation [11,20,21]. Ahn et al. described a decrease in bacterial diversity in fecal samples of CRC patients, with an increase in *Fusobacterium nucleatum* and *Porphyromonas* and a decrease in Gram-positive fiber-fermenting *Clostridia* [22]. Moreover, it has been shown that patients with colorectal tumours at an early stage (advanced adenoma) have a different microbiota composition compared with those with advanced stage tumours (definitive CRC) [19,23], suggesting that gut microbiota could participate in tumor progression.

Tjalsma et al. proposed a bacterial driver-passenger model for microbial involvement in the development of CRC, in which colonic mucosa contains bacterial species that differ in their temporal associations with developing tumours [24]. In this regard, early signs of dysbiosis in adenoma and an increased abundance of *F. nucleatum* were associated to a higher expression of pro-inflammatory cytokines in colonic tissue from CRC patients [25,26,27].

In mouse models of genetically predisposed CRC, it has been demonstrated that microbiota can elicit protumorigenic responses. For instance, Li et al. described the role of gut microbiota in the acceleration of tumor growth in APC (Min/^+^) mice by triggering the c-Jun/JNK and STAT3 signaling pathways in combination with anemia [28]. On the other hand, it has been shown that in IL-10 deficient mice, an increased microbiota-specific Th1 response exacerbated colitis, resulting in adenocarcinoma formation [29]. In germ-free mice the transfer of stool from patients with CRC enhanced intestinal cell proliferation, suggesting a promotive effect of microbiota on tumour formation [30,31,32]. 

Nevertheless, the gut microbiome is not limited only to bacteria but also includes viruses and fungal species. Many studies have reported a higher viral DNA load in tumors in comparison to normal noncancerous tissue. A number of studies have aimed to assess the potential contribution of viral infections, such as infections with human papillomaviruses, human polyomaviruses and human herpesviruses, to the risk of CRC [33,34]. 

Community-based viral shotgun NGS techniques have revealed alterations in the colon virome diversity in CRC patients. In particular CRC cohorts displayed a higher viral diversity in CRC cohorts, with enrichment in members of the genera *Orthobunyavirus*, *Inovirus* and *Tunalikevirus.* Remarkably, the last two virus genera are known to infect Gram-negative bacterial hosts, including bft-positive enterotoxigenic *Bacteroides fragilis*, *Fusobacterium nucleatum*, and pks-positive genotoxic *Escherichia coli*, which are implicated in CRC development. The fecal virome profile has been shown to be able to predict CRC status and segregate individuals at early and late stages of CRC [35]. By contrast, another study performed by Hannigan et al. did not find virome community differences in alpha diversity (richness and Shannon diversity) and beta diversity (Bray-Curtis dissimilarity) between healthy and cancerous states, although they detected strong associations between the colon virus community composition and CRC. The identified viruses were lysogenic bacteriophages belonging to the *Siphoviridae* and *Myoviridae* taxa, which can alter the composition of gut bacterial communities [36]. These authors propose a theory on how bacteriophage-bacterium dynamics may promote a novel colonization niche for cancer-associated bacteria. Thus bacteriophages could alter bacterial populations in the colon by promoting bacterial lysis, which would allow the production of biofilms by the opportunistic species anchored to the epithelium. This would favor the penetration of oncogenic bacteria in the intestinal lumen, triggering the inflammatory immune response and promoting the transformation of tumor cells [36].

Furthermore, phage therapies that exploit the co-existence of specific bacteria within cancerous tumors to induce a specific anti-tumor immune response could be used in the treatment of CRC. In fact, Zheng et al. developed a phage-guided biotic–abiotic hybrid nanosystem that could increase the chemotherapeutic potency of irinotecan against CRC cells, selectively killing the *F. nucleatum* population and allowing the butyrate-producing bacteria to increase their abundance at the same time [37]. 

On the other hand, apart from the virome, metagenome of the fungal microbiota has also been studied in CRC. The fungal genera *Phoma* and *Candida* have been detected in higher quantities in colorectal adenoma biopsies, implicating altered host-associated fungal populations in the development of CRC [38]. In another study, a fungal dysbiosis in CRC patients have been described, with enrichment in the Basidiomycota/Ascomycota ratio and the class Malasseziomycetes in CRC patients when compared with healthy controls. On the contrary, in cancer patients a decrease in the relative abundance of *Saccharomyces cerevisiae*, yeast known for its anti-inflammatory and regulatory properties of the immune system, was observed, which could make it a potential therapeutic route. Ecological analysis also revealed a higher number of co-occurring fungal intra-kingdom correlations, and more co-exclusive correlations between fungi and bacteria in CRC compared with healthy controls [39]. 

Similarly, Gao et al. observed a fungal dysbiosis in colon polyps and CRC, with an increase in the Ascomycota/Basidiomycota ratio and in the opportunistic fungi *Trichosporon* and *Malassezia*, which might favor the progression of CRC. Subsequent analysis showed a lower diversity and significant mycobiota alteration in early-stage tumors [40]. 

All these studies revealed that CRC is not only characterized by a dysbalance in the composition of gut bacteria but also by a disruption of the gut virome and mycobiome homeostasis.

### 2.2. Gut Microbiota Dysbiosis, Inflammation and CRC

Chronic inflammation has been proposed to be involved in the promotion of cancer. Thus, it is estimated that up to 20% of all tumours are preceded by chronic inflammation [41]. During carcinogenesis, inflammatory cytokines and chemokines produced by cancer cells attract immature myeloid cells or pro-inflammatory helper T cells. This pro-tumorigenic microenvironment is characterized by the synthesis of growth and angiogenic factors and tissue remodelling enzymes, and the suppression of antitumor T-cell responses [42], favouring tumour progression. Gut microbiota dysbiosis and increased intestinal permeability are highly associated to colon inflammation, which could be a key factor for the initiation and/or progression of CRC [43]. Thus, when intestinal permeability is increased, the lipopolysaccharides of the outer membranes of some types of bacteria penetrate the host organism, which induces the immune system to secrete cytokines and start a cascade of reactions that ultimately leads to inflammation. Local inflammation contributes to tumor progression through protumorigenic cytokines and chemokines that act as growth factors and promote angiogenesis [42]. In mouse models it has been recently demonstrated that the development of polyps was associated with defects in the colon barrier integrity, bacterial invasion, and an increased expression of several inflammatory factors such as IL-17, *Cxcl2*, *Tnf-α*, and IL-1. Moreover, alterations in the intestinal barrier allowed microbes to induce local inflammation, promoting polyp formation and cancer development in mice [44]. Hu et al. demonstrated that aberrant inflammasome-induced microbiota plays a critical role in CRC development, where mice deficient in the NOD-like receptor family pyrin domain containing 6 (NLRP6) inflammasome exhibited enhanced inflammation-induced CRC formation [45].

### 2.3. Pathogenic Bacteria and CRC

In addition to a shift in the microbiota composition, pathogenic bacterial species may also have a role in the development of CRC. There are different pathogenic microbes associated to the promotion of CRC, including several *Bacteroides* species (*B. vulgatus* and *B. stercoris*), *Bifidobacterium* species (*B. longun* and *B. angulatum*), *Eubacterium* species (*E. rectale* 1 and 2, *E. elignes* 1 and 2, and *E. cylindroides*), *Ruminococus* species (*R. torques*, *R. albus*, and *R. gnavus*), *Streptococo hansenii*, *Fusobacterium prausnitzi*, and *Peptoestreptococo productus* 1 [46]. All these microbes may drive CRC tumorigenesis by inducing proliferation of the epithelial cells, producing damage in the epithelial barrier, and causing inflammation. In addition, different toxins may damage DNA inducing a protumorigenic effect. For instance, *Bacteroides fragilis* toxin is known to activate Wnt and NF-kB signaling pathways and enhance epithelial release of pro-inflammatory molecules [47,48], *E. coli* toxin (colibactin toxin) causes DNA crosslinks and double strand DNA breaks [49], and Salmonella protein AvrA has recently been shown to induce β-catenin signaling and enhance colonic tumorigenesis by activating STAT3 pathway in a colon cancer mouse model [48]. Similarly, *F. nucleatum* has emerged as a potential candidate for CRC predisposition, because of its ability to bind to E-cadherin on the surface of colon cells through FadA adhesion, leading to the activation of Wnt/B-catenin signaling and the production of an inflammatory and oncogenic response [50]. Fap2, another adhesin from *F. nucleatum*, is able to bind to the inhibitory immune receptor TIGIT (T cell immune receptor with Ig and ITIM domains) and alter the function of natural killer cells and tumor infiltrating lymphocytes [51]. *F. nucleatum* has also been associated with resistance to the CRC chemotherapy agent oxaliplatin by inducing authophagy via Toll-like receptor 4 [51]. 

## 3. Gut Microbiota-Derived Metabolites and CRC

The gut microbiota produces different metabolites after anaerobic fermentation of exogenous undigested dietary components. These metabolites interact with the epithelial cells of the mucosal interface, influencing immune responses and the potential development of different diseases. The gut microbiota derived metabolites with pro-carcinogenic effects include products of protein fermentation such as polyamines [52]. Remarkably, a recent metagenomic analysis reported that the CRC-associated microbiome showed an association with alterations in polyamine metabolism [53], indicating that these metabolites could be particularly important in CRC development and progression. On the other hand, CRC has been associated to alterations in the metabolism of SCFAs [8,9], which have been shown to exhibit potential anti-carcinogenic effects in cellular and animal models of colon cancer. 

### 3.1. Gut Microbiota-Derived Polyamines and CRC

Polyamines are aliphatic amines essential for normal cell growth. It is widely accepted that polyamine metabolism is frequently dysregulated in cancer, including CRC [54,55]. In colon cancer, ornithine decarboxylase (ODC), the key enzyme of the polyamine biosynthetic pathway, is expressed at higher levels in tumor tissue than in adjacent normal mucosa [56,57], suggesting that increased polyamine production could be involved in the tumorigenesis of CRC. The restriction of polyamine availability by alpha-difluoromethylornithine (DFMO) treatment, a chemical inhibitor of ODC, in combination with non-steroidal anti-inflammatory drugs has been shown to exhibit promising effects as therapeutic option for colorectal adenoma incidence [58,59]. A limitation of the monotherapy with DFMO, however, is that tumor cells can replace endogenously synthetized polyamines by taking extracellular polyamines from the colon lumen. This is particularly important in the case of CRC, which is surrounded by intestinal bacteria that are able to produce high levels of polyamines [60,61]. Remarkably, a recent metagenomic analysis has established that the CRC-associated microbiome showed an association with the conversion of amino acids to polyamines (e.g., L-arginine and L-ornithine degradation to putrescine) [53]. In mice, the administration of antibiotics enhanced the cytostatic effect of DFMO on tumor cells [62,63], suggesting that reduction of bacterial polyamine biosynthesis together with the inhibition of the polyamine biosynthesis route could be considered as an anti-tumoral strategy. Another promising strategy to limit the availability of polyamines in the tumor could be the combination of DFMO and the polyamine transport inhibitor AMXT 1501, which has shown to be effective in mouse models of neuroblastoma [64].

On the other hand, a metabolomics screen comparing paired colon cancer and normal tissue samples from patients with CRC revealed that bacteria biofilm formation, even in the normal colon tissue, was associated with increased colonic epithelial cell proliferation and host-enhanced polyamine metabolism [65]. In addition, bacteria-generated polyamines in biofilms may contribute to the inflammation and proliferation of colon cancer [58]. Following antibiotic treatment, resected colorectal cancer tissues harbored disrupted bacterial biofilms and lowered N^1^, N^12^-diacetylspermine tissue concentrations compare to biofilm-negative colon cancer tissues, suggesting that gut microbes can induce an increase of host generated N^1^, N^12^-diacetylspermine [66] (Figure 1).

### 3.2. Short Chain Fatty Acid Metabolism and CRC

SCFAs, especially butyrate, propionate and acetate, are products of the fermentation of dietary fiber by anaerobic gut microbiota with an essential role in the health of colonic mucosa through the modulation of the local immune response and the protection of the intestinal barrier. Recent studies have reported lower levels of butyrate-producing bacteria in CRC patients [18,67]. In addition, metabolomic analyses have described significant perturbations of SCFA metabolism in CRC compared to adjacent mucosa [68]. Butyrate has been shown to be able to induce IL-18 production in intestinal epithelial cells by activating GPR109a receptor, which stimulates the mucosal tissue repair through the regulation of the production and availability of IL-22 [69]. Remarkably, in mice the absence of IL-18 has been associated with gut microbiota dysbiosis, alterations of the inflammatory response, and a dysregulation of the homeostatic and mucosal repair [70,71], resulting in increased susceptibility to carcinogenesis. In fact, some experiments with mice that are unable to respond to IL-18 have shown a high incidence of intestinal dysbiosis and elevated susceptibility of chemically induced CRC carcinogenesis [69,72,73]. In addition, butyrate can induce the expansion of T reg lymphocytes to regulate the local immune response and suppressing colonic inflammation and carcinogenesis [74] (Figure 1).

## 4. Antibiotic-Microbiome Link and CRC Risk

The use of antibiotics generally has broad effects on the gut microbiota and indirectly affects CRC progression. The suppression of microbiota by antibiotics has been related to a decrease in crypt height and heme-induced colorectal carcinogenesis in rats [75]. Dick et al. described in a nested case–control study that the use of antibiotics with both anti-anaerobic or anti-aerobic activity (such as penicillins and quinolones) was associated with a dose-dependent increased risk of CRC development [76]. Interestingly, another nested case-control study in UK has shown that bacterial or fungal outgrowth after multiple penicillin treatments slightly increases the risk of CRC development [77]. ZacKular et al. demonstrated that manipulation of the gut microbiota with different antibiotic cocktails during the onset of inflammation can significantly decrease tumorigenesis in mice [78]. Bullman et al. showed that the treatment with metronidazole of mice xenografted with CRC decreased both the load of *F. nucleatum* and the growth of the tumor [79]. In an azoxymethane (AOM)/dextran sodium sulfate (DSS)-induced CRC murine model the alteration of gut microbiota using antibiotics attenuated colon tumorigenesis, but only when gut microbial changes were maintained throughout the entire period of inflammation [80]. Recently, Ma et al. described that the alterations of gut microbiota after antibiotic use could contribute to the long-term dysregulation of host immune homeostasis and affect CRC pathogenesis [81]. In another meta-analysis, Wang et al. suggested that a higher number of antibiotic prescriptions were associated with a higher risk of CRC. By contrast, they described that the risk of rectal cancer was inversely associated with antibiotic exposure, possibly due to the differences in the composition of gut microbiota between colon and rectum [82].

On the other hand, the use of antibiotics in early childhood has been associated with increased colonic adenoma formation (a precursor lesion to CRC) in later life, suggesting that a dysbiotic microbiota is acquired and hold over a longer period of time [83,84,85]. In a recent study based on the Clinical Practice Research Datalink (CPRD), the use of oral antibiotics with anti-anaerobic activity has been associated with increased CRC risk in a dose-dependent fashion in the UK population, although the effects differed depending on the anatomical location, being greatest in the proximal colon [86]. In this regard, a systematic review and meta-analysis of observational studies was performed to assess whether the use of antibiotics was associated with the development of pre-cancerous or cancerous lesions in adults [87]. In this review the authors only found a weak association between cumulative antibiotic use and risk of CRC. These results could be explained by confounding factors within the studies, such as heterogeneity in how antibiotic exposure was registered, the variability in the route and setting of antibiotic exposure among studies, the relatively short time between antibiotic exposure and the development of CRC in the majority of studies, and that none of the included studies tried to differentiate microbiome-associated events between initiation of CRC as polyp prevalence and progression through more advanced stages [87]. 

Another recent work, Armstrong et al. showed that patients prescribed antibiotics in up to 15 years preceding diagnosis were associated with a higher risk of CRC [88]. 

On the other hand, the use of antibiotics often leads to dysbiosis, facilitating the acquisition of drug resistance. In this context, Yuan et al. found that antibiotic treatment-induced gut microbiota dysbiosis decreased the therapeutic efficacy of 5-fluorouracil (5-FU) for tumor treatment [89]. Remarkably, antibiotic use before (but not following) the start of 5FU-based chemotherapy has been associated with worse progression-free and overall survival among patients with metastatic colorectal cancer [90]. 

In addition, the disruption of microbiota in MC38 colon carcinoma-grafted mice with a broad-spectrum antibiotics impaired tumor response to anti-CTLA4 immunotherapies [91]. Nevertheless, the efficacy of anti-CTLA4 treatment in antibiotic-treated MC38-grafted mice could be rescued by colonizing mice with *B. fragilis*, immunizing with low dose of a recombinant BFT-2 enterotoxin (a major virulence factor of *B. fragilis*), or performing adoptive transfer with *B. fragilis*-specific T cells [92]. 

## 5. Dietary Mediators of CRC Carcinogenesis

### 5.1. Dietary Fiber

Dietary fiber has been shown to beneficially affect metabolic activities in the gastrointestinal tract [93,94]. In the observational EPIC study, Bingham et al. found that dietary fiber was inversely associated with the incidence of large bowel cancer, although no significant differences were observed between various food sources of dietary fiber intake on the protection against CRC [95]. In another large prospective cohort study, the protective effect of whole-grain consumption was associated with a slight reduction in the risk of developing CRC [96]. In a prospective case-control study nested within seven UK cohort studies using food intake questionnaires, both the intake of absolute fiber as well as the fiber intake density were inversely associated with the risk of colorectal and colon cancers in both age-adjusted models and multivariable models adjusted for age, anthropomorphic and socioeconomic factors, dietary intake of folate, alcohol consumption, and energy intake [97]. The protective effect of dietary fiber remained evident in the 11-year follow-up of the EPIC study, being the total dietary fiber intake still inversely associated with colorectal cancer [98]. Similarly, a prospective study in the Scandinavian HELGA cohort showed that the intake of dietary fiber (especially from cereals) was associated with a reduction in the incidence of CRC [99]. Moreover, a higher intake of dietary fiber and whole grains after CRC diagnosis has been associated with better survival rates [100] 

Moen et al. compared the effects of several dietary interventions (inulin, cellulose or brewers spent grain) in AOM -treated A/J Min/+ mice, finding that the mice fed with inulin displayed lower incidence of colonic tumorigenesis and a distinct cecal microbiota profile associated with low colonic tumor load [101]. 

On the other hand, Mehta et al. found that diets rich in whole grains and dietary fiber were associated with a lower risk of developing *F. nucleatum*-positive colorectal cancer but not *F. nucleatum*-negative CRC, supporting a potential role for intestinal microbiota in mediating the association between diet and the development of colorectal neoplasms [102]. 

More recently, the fermentation of soluble fibers such as lignan and β-glucan to SCFAs by gut microbiota also plays a critical role in cancer prevention. The ingestion of dietary fiber has been associated with the presence of fecal butyrate-producing bacteria [103,104]. Remarkably, lower fecal SCFA levels as a consequence of a lower dietary fiber intake and lower prevalence of *Clostridium*, *Roseburia*, and *Eubacterium* spp. were found in CRC risk subjects compared to healthy individuals [104]. In agreement, a dietary intervention consisting in a higher intake of dietary fibers in African Americans increased saccharolytic fermentation and butyrogenesis, and suppressed secondary bile acid synthesis, resulting in the reduction of biomarkers of colon cancer risk [105]. In line with this work, another study using a gnotobiotic mouse models demonstrated that dietary fiber protected against colorectal tumorigenesis in a microbiota- and butyrate-dependent manner via inhibition of histone deacetylase activity [106]. In addition, high fiber diets given to mouse models of polyposis also produced a significant increase of SCFA-producing bacteria and ameliorated polyposis [107]. 

The possible mechanism that could explain the role of dietary fiber in CRC prevention could be that fiber reduces concentrations of intestinal carcinogens due to the reduction of intestinal transit time and increased faecal bulk, which would lessen the potential for faecal mutagens to interact with the colon mucosa [83]. In addition, the increase bacterial fermentation of resistant starch to SCFAs (especially butyrate) has been shown to lower fecal pH in the colon, and this pH reduction can inhibit pathogenic organism proliferation and DNA damage induction, and enhance apoptosis and prevent proliferation of cancer cells [108,109,110]. On the other hand, it has been decribed that long-term fiber dominant diet may increase the density of *Firmicutes*, which may have immune modulatory and anti-inflammatory effects in the host [111,112] (Figure 1). 

Altogether, a higher fiber intake could not only prevent the disturbances in the community structure and function of the gut microbiota, but could also stimulate the production of bacterial metabolites with anti-CRC activity such as butyrate (see below). Moreover, the intake of fiber after CRC diagnosis has been associated with better survival rates. Nevertheless, more clinical and preclinical studies are necessary to establish the most appropriate conditions (dose and duration) of these dietary interventions involving high-fiber intake to prevent CRC (Table 1).

### 5.2. Diets Rich in Polyunsaturated Fatty Acids

Diverse studies have recently defined an impact of dietary omega-3 polyunsaturated fatty acids (PUFAs) on the gut microbiota [113]. In particular, the supplementation of PUFAs has been associated with a decrease in *Faecalibacterium* and an increase of *Bacteroidetes* and butyrate-producing bacteria [114]. In addition, PUFAs are able to reduce intestinal microbial dysbiosis by increasing the proportions of beneficial bacteria and decreasing the proportions of pathogenic bacteria in the gastrointestinal tract [115]. 

PUFAs have been extensively studied due to their role in their protective effect from CRC carcinogenesis, mainly through mechanisms that regulate differentiation and apoptosis of the colonocytes [116,117,118,119]. For instance, in C57BL/6J mice bearing azoxymethane-dextran sulfate sodium–induced CRC, the relative abundance in the gut of beneficial bacteria such as *Lactobacillus* increased after eicosapentaenoic acid treatment, in parallel with a reduction in the size of colorectal tumors, a decrease in the number of proliferating cells, and an increase of apoptotic cells within the tumors [120]. These PUFAs could also alter the cell cycle components, act on the immune system and modulate CRC-related genes expression [121]. The protective role of PUFAs in colorectal carcinogenesis prevention may also relate to the decreased risk of microsatellite instability (MSI) and the enhancement of DNA repair systems mismatch pathways [122]. 

On the other hand, several randomized control trials have reported that PUFAs are often subjected to peroxidation, process by which free radicals are frequently generated [123]. In addition, Yang et al. found that the PUFA composition is different between normal and cancerous tissues in the same CRC patient, suggesting that the metabolism of PUFAs might play a significant role in the evolution of inflammation driven tumorigenesis in the CRC [124]. 

In rats exposed to azoxymethane, a potent carcinogen used to induce colon cancer in animal models, the consumption of dietary fish oil (which is rich in omega-3 PUFAs) led to a lower rate of CRC adenocarcinoma incidence [125]. Song et al. reported that high marine ω-3 PUFA intake after CRC diagnosis is associated with a lower risk of CRC-specific mortality, indicating that an elevated consumption of marine ω-3 PUFAs after diagnosis may confer additional benefits to patients with CRC [126]. Furthermore, other investigations have demonstrated that in animals with carcinogen-induced CRC tumors fed with a diet of fish oil plus pectin had increased colonocyte apoptosis compared with those fed with corn oil cellulose as control diet [127]. Moreover, it has been recently shown that docosahexaenoic acid in combination with butyrate enhances mitochondrial lipid oxidation and reduces mitochondrial membrane potential, contributing to the induction of apoptosis in colonocytes [109,128]. A recent clinical trial has shown a CRC incidence reduction of 22% among pesco-vegetarian subjects compared with non-vegetarian individuals [129]. Finally, a very recent study performed by Aglago et al. analyzed the association between fish consumption and dietary and circulating levels of PUFAs with CRC incidence using data from the EPIC cohort. They found that the regular intake of fish at recommended levels was associated with a lower risk of CRC, possibly through the exposure to high PUFA content [130]. 

Several in vivo studies have described that PUFAs can reduce 5-FU-related toxicity and potentiate 5-FU anti-cancer activity though the reduction of tumor burden and DNA damage and the increase of apoptosis [131,132,133]. A recently study in rat models showed that combining PUFAs with 5-FU and irinotecan could help restore lipid stocks, thus potentially limiting 5-FU-associated side effects [134]. Cai et al. showed that PUFAs have the potential to radio-sensitise HT29 colon cancer cells, possible due to an increase in lipid peroxidation products within the cells [135]. Granci et al. reported an increase in apoptosis in colon cancer cells when combining 5-FU, oxaliplatin, and irinotecan with a fish oil emulsion with PUFAs [136] (Figure 2). 

A recent double-blind, randomized, placebo-controlled trial investigated the effect of the combination treatment of PUFA and a probiotic supplement on the tolerability of capecitabine/oxaliplatin chemotherapy and on inflammatory markers in CRC patients, finding an improved overall quality of life and reduced chemotherapy-induced symptoms such as diarrhea and fatigue in these study subjects [137]. 

Then, PUFAs have a protective effect from CRC carcinogenesis and in combination with chemotherapeutic agents could be an effective approach to the treatment of CRC patients (Table 1). 

### 5.3. Bioactive Polyphenols

Most fruits and vegetables contain phytochemicals with anti-microbial and anti-inflammatory properties [138]. Phytochemicals are able to maintain the balance of the gut microbiota and exhibit anti-tumoral properties (e.g., decrease cell proliferation and stimulate apoptosis of cancer cells, inhibit angiogenesis and delay metastasis) [139]. Polyphenols are a structural class of phytochemicals with multiple phenolic units that are found at high concentrations in coffee, tea, wine, fruits, vegetables and whole grains [140,141]. Because polyphenols are poorly absorbed in the small intestine they usually tend to accumulate in the colon, where they can be hydrolyzed by the enzymatic activities of the gut microbial community into lower molecular-weight bioactive compounds before absorption [142,143,144]. Moreover, polyphenols present in the colon have been found to significantly alter the gut microbiota, particularly by suppressing the growth of *Clostridium* and *Bacteroides* species [145,146,147].

On the other hand, red wine polyphenols have been linked to CRC prevention by their capacity of inhibiting the growth of pathogenic bacterial species such as *F. nucleatum* and *P. gingivalis* [148], as well as the adhesion to oral cells [149]. 

Several studies have described the effects of certain polyphenolic compounds in CRC prevention and treatment both in vivo and in vitro. Quercetin, a flavonol present at high concentrations in certain vegetables and fruits such as onions or apples, has been shown to exert some anticancer effects in colon cancer cells, mainly by inhibiting cell proliferation and inducing apoptosis [150]. Anthocyanin-rich tropical fruits such as Cocoplum (*Chrysobalanus icaco* L.) have also demonstrated anti-inflammatory activity (through the reduction of TNF-α, IL-1β, IL-6, and NF-κB expression levels) and pro-oxidant effect in the human colorectal adenocarcinoma cell line HT29 [136,151]. In HCT116 colon cancer cells, the activity of apigenin was correlated with a blockage in cell cycle progression, induction of apoptosis and inhibition of autophagy [152]. The antitumor effects of several polyphenols present in high amounts in blueberries, red grapes and cocoa (such as anthocyanins and tannins) have been related to their capability of inducing adaptive immune cells to target tumor cells in preclinical models [153,154,155]. In addition, the combination of curcumin and resveratrol has been shown to be highly effective in inhibiting the growth of colon cancer cell both in vitro and in vivo [156]. In some clinical studies, the intake of flavonols and flavan-3-ol monomers has been associated with a decreased risk in colorectal cancer [157]. However, the association between the regular intake of either total flavonoids or any flavonoid subclass and CRC risk and tumor subsites could not be corroborated in other human cohort studies [158,159]. 

Emerging evidence suggests that the combination of conventional chemotherapy treatment for CRC with some natural dietary polyphenols can significantly enhance the chemotherapeutic effect. For instance, quercetin has been tested in vitro in combination with 5-FU in CO115 human colon carcinoma cells and HCT15 colorectal adenocarcinoma cells increasing apoptosis levels in CO115 cell line, in a synergistic manner, but as an additive effect in HCT15 cells [160]. Also, the combination of 5-FU against a colon adenocarcinoma cell line treatment with phenolic acid rich-extracts such as Gelam honey and ginger (*Zingiber officinale*) enhanced the anticancer activity of 5-FU [140,161]. On the other hand, Montrose et al. demonstrated in DSS-mice that the chemopreventive effect of black raspberries was mediated by the downregulation of the expression of pro-inflammatory cytokines (TNF-α and IL-1β) and the decrease of COX-2 and plasma prostaglandin E2 levels [162]. On the other hand, the chemopreventive effect of curcumin through the reduction of colonic tumor burden due to the maintenance of a high microbial diversity was proven in IL-10-deficient mouse [163]. Moreover, curcumin is able to enhance chemosensitization to 5-FU-based chemotherapy by targeting cancer stem cell subpopulations that could be responsible for tumor relapse and resistance to conventional therapies [164]. The effect of 5-FU also increased in combination with resveratrol due to its chemosensitizing properties [165]. Moreover, resveratrol could be used to overcome drug resistance in combination with other chemotherapeutic drugs, due to its ability to downregulate multidrug resistant protein 1 by preventing the activation of NF-κB signalling and suppressing cAMP-responsive element transcriptional activity [166] (Figure 2).

These findings suggest that polyphenols and their derived microbial metabolites could be used as a complementary therapy not only for CRC prevention, but also in potentiating the efficacy of chemotherapy against CRC (Table 1). 

### 5.4. Probiotics

Probiotics are live microorganisms that contribute to the health benefit of the patients and are able to inhibit CRC through different mechanisms (Figure 2). Several studies have suggested that regular consumption of probiotics may improve the diversity and richness profile of the intestinal microbiota, downregulate chronic inflammation, and reduce the production of carcinogenic compounds during intestinal dysbiosis [167,168].

Hatakka et al. demonstrated that the consumption of certain strains of probiotic bacteria can reduce the activity of intestinal enzymes that can convert aromatic hydrocarbons and amines in active carcinogens and prevent colon cancer [169]. The peptidoglycan, polysaccharide and secreted glycoproteins on the surface of probiotic bacteria, combined with carcinogenic mutagens could be responsible for biotransformation aiming to detoxification [170].

On the other hand, probiotics can also regulate the immune system response through the activation of phagocytes to eliminate cancer cells in their early stages of development, contributing to the maintenance of the immune-vigilance state [171,172]. For instance, the probiotic strain *Bifidobacterium animalis* subsp. *Lactis* can produce mycosporin-like amino acids, which are able to modulate host immunity by regulating the proliferation and differentiation of intestinal epithelial cells, macrophages and lymphocytes and the production of cytokines [173]. Nevertheless, not all probiotics are able to regulate the immune system and to prevent the development of CRC. To induce immunostimulation on the host, both a dosage of around 10^9^ CFU/day and an intestinal transit time between 48 and 72 h are necessary [168]. 

Probiotics are also known to stimulate the production of a variety of compounds that improve the intestinal barrier function. The perioperative administration of CRC patients with a probiotic consisting in a mixture of *Lactobacillus plantarum*, *Lactobacillus acidophilus* and *Bifidobacterium Longum* increased the expression of mucosal tight junction proteins, improved the integrity of gut mucosal barrier and reduced enteropathogenic bacteria, resulting in decreased infectious complications after colorectomy [174]. 

In one randomized, double-blind, placebo-controlled trial, patients with colon cancer and polypectomized patients, Rafter et al. demonstrated that oral treatment with a probiotic mixture of *Lactobacillus rhamnosus* and *Bifidobacterium Breve* was able to induce changes in gut microbiota, reduce several cancer biomarkers such as colorectal proliferation, and improve intestinal epithelial barrier permeability [175]. Interestingly, when the bacterial probiotic strains *L. acidophilus NCFM* and *B. animalis* subsp. *lactis Bl-04* were used in a prospective intervention study with CRC patients, it was observed that the patients with colon cancer that received probiotics had a unique profile of bacterial populations in their gut microbiota, which was mainly characterized by an increased abundance of butyrate-producing species in tumor, mucosa and fecal samples [176]. Other probiotic strains derived from *Bifidobacterium* have been shown to restore the equilibrium of the gut dysbiosis in patients with CRC [177]. 

The consumption of probiotics has been also associated with the induction of a proapoptotic activity in cancer cells of human CRC patients. Wan et al. found that the consumption of *Lactobacillus Delbrueckii* increased the expression of caspase-3, leading to apoptosis of human colon cancer cells [178]. Furthermore, Konishi et al. described that *L. casei* strain *ATCC 334* produced ferrichrome, a molecule that inhibits the progression of colon cancer by inducing the apoptosis of cancer cells via c-Jun N-terminal kinase pathway [179]. 

In addition, several studies have described a relationship between gut microbiota and the efficacy and/or toxicity of both chemotherapies and immunotherapies [180,181]. Probiotics has been also shown to affect the response to immunotherapy by systemic priming and regulation of different myeloid-derived cell functions in the tumor microenvironment. Tumor-infiltrating myeloid cells appear to be primed by bacterial LPS through the TLR4 receptor for responsiveness to the TLR9 ligand CpG-ODN [181]. In this regard, Chang et al. described that *L. casei* variety *rhamnosus* (*LCR35*) attenuated 5-FU/oxaliplatin-induced intestinal mucositis in CRC-bearing mice [182]. Moreover, recent studies have described that the combination of PD-L1 inhibitors and oral therapy with bifidobacteria had a synergistic inhibitory effect on tumor growth compared with the effect of either intervention alone [183,184,185]. In CRC patients undergoing chemotherapy the supplementation with *L. rhamnosus* decreased the frequency of diarrhea and abdominal distress and avoided the dose reduction caused by intestinal toxicity compared to patients who received placebo [186]. 

Finally, probiotics have been studied in the setting of radiation therapy used in the treatment of CRC patients. Previous studies have demonstrated that intestinal bacteria can repair injuries and reduce the incidence and severity of diarrea and bowel movements induced by the radiation therapy [187]. 

Taken together, the probiotic interventions could be therapeutically used to target the gut dysbiosis frequently observed in CRC patients, due to their beneficial effects on the immune system and the intestinal barrier function, as well as, their antitumoral properties by releasing metabolites that are able to get rid of potential carcinogens. Moreover, the modification of the composition of CRC microbiome with probiotics might enhance the effectiveness to cancer chemotherapy and immunotherapies and the reduction of toxicity associated to radiation therapy. Noteworthy, not all probiotic strains showed anti-CRC effects and their beneficial impact depends on the bacterial strain, the dosage, the duration of intervention and the intestinal transit time. Further investigations are therefore necessary to clarify the action mechanism and the potential of probiotics in CRC prevention (Table 1).

## 6. Conclusions

Different animal and human studies have revealed that the microbial composition has been altered in precancerous colorectal lesions and in CRC. Moreover, a dysbiosis in gut microbiota has been found in CRC patients compared with healthy controls, with enrichment in pro-inflammatory opportunistic pathogens and a decrease in butyrate-producing bacteria. The proposed mechanisms by which the gut microbiota dysbiosis could participate in colorectal carcinogenesis are the impairment of the intestinal epithelial barrier function, the triggering of proinflammatory responses, the biosynthesis of genotoxins that can interfere with cell cycle regulation, and the production of toxic metabolites by pathogenic bacteria. Moreover, some lysogenic bacteriophages from gut virome could alter bacterial populations (by promoting bacterial lysis) in the colon, which could indirectly result in tumor progression. In addition, ecological analyses revealed synergistic intrafungal and antagonistic bacterial–fungal interactions in colorectal carcinogenesis, suggesting that gut mycobiota may also contribute to colorectal tumorigenesis.

In patients with CRC bacteria biofilm formation was associated with a host-enhanced polyamine metabolism, which may significantly contribute to the inflammation and cellular proliferation of colon cancer cells. Antibiotics were associated with CRC risk, but the effect depends on anatomical location and the type of antibiotics. On the other hand, previous epidemiological and clinical research studies have demonstrated that diet plays an important role in the promotion or inhibition of CRC, and gut microbiota is one of the most important links between them. High-fiber diets can significantly reduce the risk of CRC development. Soluble fiber is fermented into SCFAs by bacteria in the large intestine, and SCFAs (especially butyrate) has been shown to exhibit potential anti-carcinogenic effects in in vivo colon cancer models by modulating the local immune response and the protection of the intestinal barrier. Moreover, a high-fiber intake can increase the number of butyrate-producing bacteria in the gut. In addition, supplementation with PUFAs, polyphenols and probiotics could be used as therapeutical approaches for the reduction of CRC risk in a primary prevention setting, and it may also be used as adjuvants to conventional treatment for CRC, given the fact that the intestinal microbiota may modulate and enhance response to cancer therapy and reduce toxicity. 

Thus, taking all this evidence together, gut microbiota should be considered as a key factor that can contribute to both the initiation and development of CRC. In addition, the dietary modulation of cancer-associated microbiome, through the intake of dietary components able to avoid dysbiosis and intestinal inflammation or to help modulate response to cancer therapy, could be an efficient strategy to prevent the development and progression of CRC and improve the efficacy of therapy.

## Figures and Tables

**Figure 1 cancers-12-01406-f001:**
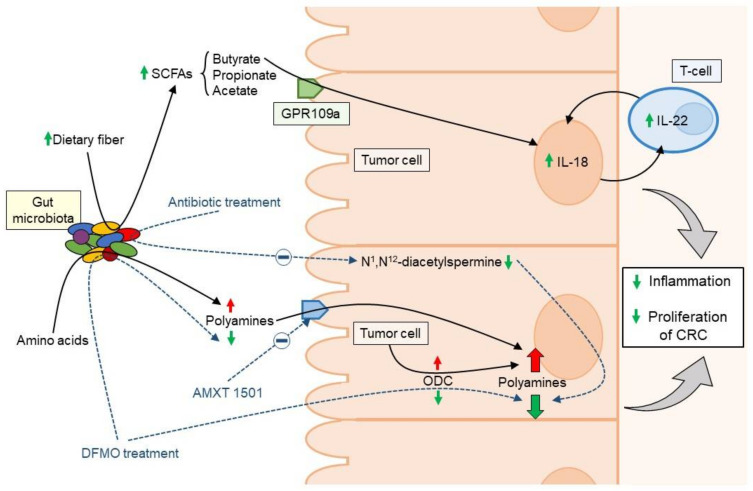
Mechanisms of action of polyamines and SCFAs (microbiota-derived metabolites) in the inflammation and cellular proliferation of colon cancer cells. SCFAs: short-chain fatty acids; ODC: ornitina descarboxilasa; DFMO: alpha-difluoromethylornithine; AMXT 1501: polyamine transport inhibitor; GPR109a: G-protein–coupled receptors; IL: interleukin.

**Figure 2 cancers-12-01406-f002:**
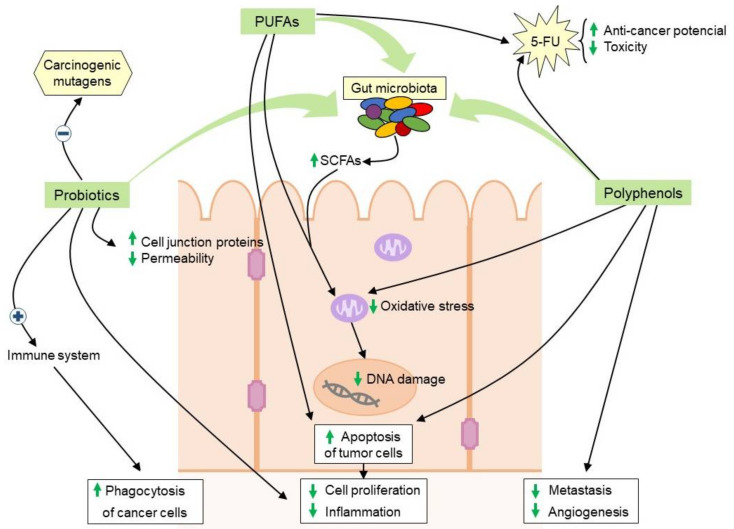
Beneficial effects of dietary supplementation with PUFAs, polyphenols and probiotics on the intestinal microbiota and colon cells for the reduction of CRC risk or to enhance the response to cancer therapy when are used as adjuvant to conventional treatment. PUFAs: omega-3 polyunsaturated fatty acids; 5-FU: 5-fluorouracil; SCFAs: short-chain fatty acids.

**Table 1 cancers-12-01406-t001:** Interactions between dietary mediators, gut microbiota and CRC.

Study (Reference)	Dietary Mediator	Type of Study	Species	Most Relevant Results
***Dietary Fiber***				
Lattimer et al. 2010 [93])	Dietary Fiber (Arabinoxylan, Inulin, β-glucan, Pectin, Bran, Cellulose, Resistant Starch)	In vivo	Human	↑ Excretion of bile acids, ↑ Production of fecal SCFAs ↑ Antioxidants ↓ Cancer prevalence
Zeng et al. 2014 [94]	Dietary Fiber	In vivo	Human	↓ Fecal pH in the colon ↑ SCFA-producing gut bacteria ↑ Apoptosis of colon cancer cells ↓ Chronic inflammatory process and migration/ invasion of colon cancer cells
Deehan et al. 2020 [103]	Dietary Fiber	In vivo	Human	Modulation of the colon microbiota ↑ Saccharolytic fermentation ↑ Production of fecal SCFAs
Chen et al. 2013 [104]	Dietary Fiber	In vivo	Human	↑ Production of SCFAs by healthy gut microbiota, ↓ Risk of advanced colorectal adenoma.
Burkitt et al. 1993 [108]; Bergman et al. 1990 [109]; Hamer et al. 2008 [110]	Dietary Fiber	In vivo	Human/ Mouse	↑ Production of fecal SCFAs (especially butyrate) ↓ Fecal pH in the colon, ↓ Pathogenic organism proliferation ↓ DNA damage induction ↑ Apoptosis of colon cancer cells ↓ Proliferation of colon cancer cells.
Fung et al. 2012 [111]; Neish et al. 2009 [112]	Long-term dietary fiber intake	In vivo	Human	↑ Abundance of *Firmicutes* abundance ↑ Immune modulatory and anti-inflammatory effects in the host
Bingham et al. 2003 [95]	Dietary Fiber	In vivo	Human	↑ Total dietary fiber intake ↓ Risk of CRC
Schatzkin et al. 2007 [96]	Dietary Fiber (whole grains)	In vivo	Human	↑ Whole grain food consumption ↓ Risk of CRC (modest)
Dahm et al. 2010 [97]	Dietary Fiber	In vivo	Human	↑ Fiber intake ↓Risk of CRC
Hansen et al. 2012 [99]	Dietary Fiber (cereals)	In vivo	Human	↑ Total dietary fiber ↓ Risk of CRC
Song M et al. 2018 [100]	Dietary Fiber (whole grains)	In vivo	Human	↑ Survival rates of non-metastatic CRC
Moen et al. 2016 [101]	Dietary Fiber (inulin, cellulose, brewers spent grain)	In vivo	Mouse	Inulin intake change cecal microbiota ↓ Colonic tumorigenesis
Mehta et al. 2017 [102]	Dietary Fiber (whole grains)	In vivo	Human	↓ Risk of developing *Fusobacterium nucleatum*-positive CRC
O’Keefe et al. 2015 [105]	Dietary fiber and fat	In vivo	Human	↑ Saccharolytic fermentation ↑ Butyrogenesis ↓ Secondary bile acid synthesis ↓ Biomarkers of colon cancer risk
Donohoe et al. 2014 [106]	Dietary Fiber	In vivo	Mouse	↑ Microbial fiber fermentation ↑ Butyrate production ↑ Protection against colorectal tumorigenesis.
Bishehsari et al. 2018 [107]	Dietary Fiber	In vivo	Mouse	↑ SCFA-producing bacteria, ↓ Gut microbiota dysbiosis ↓ Polyposis incidence
***Diets rich in polyunsaturated fatty acids***				
Costantini et al. 2017 [114]	n-3 PUFAs	In vivo	Human/ Mouse	↓ Relative abundance of *Faecalibacterium* ↑ Bacteroidetes and butyrate-producing bacteria (*Lachnospiraceae* family)
Cho et al. 2014 [116]	n-3 Fatty Acid Docosahexaenoic Acid and Butyrate	In vitro	Human	Epigenetic alterations (methylation of proapoptotic genes) ↑ Apoptosis of colon cancer cells
Chapkin et al. 2014 [117]	n-3 PUFAs	In vivo	Human/ Mouse	Alterations in the plasma membrane of colon cancer cells Epigenetic alterations ↑ Risk of developing CRC.
Triff et al. 2015 [118]	n-3 PUFA & Fiber	In vivo	Human/ Mouse	Regulation of nuclear receptor transcriptional activity ↓ Inflammatory cytokines ↑ Chemoprotection.
Hong et al. 2015 [119]	Fish oil & Butyrate	In vivo	Rat	↑ Apoptosis of colon cancer cells ↓ Proliferation of colon cancer cells ↑ Protection against CRC
Lee et al. 2017 [121]	Fish oil & Butyrate	In vivo	Human	Modulation of CRC-related gene expression ↓ Inflammation ↑ Apoptosis of colon cancer cells
Chang et al. 1998 [125]	Fish oil & Fiber (pectin, cellulose)	In vivo	Rat	↑ Apoptosis of colon cancer cells ↓ Proliferation of colon cancer cells ↓ Rate of CRC adenocarcinoma incidence
Cho et al. 2012 [127]	Fish oil & Pectin	In vivo	Rat	↑ Apoptosis of colonocytes ↑ Chemoprotective capacity
Ng et al. 2005 [128]	Docosahexaenoic acid (DHA, 22:6 n-3) & butyrate	In vitro	Human	↑ Mitochondrial lipid oxidation ↓ Mitochondrial membrane potential ↑ Apoptosis of colonocytes ↑ Chemoprotective effects
Sofi et al. 2019 [129]	Comparison of Meat-Based vs Pesco-Vegetarian Diets	In vivo	Human	Positive effect of pesco-vegetarian diet on gut microbiota ↓ Risk of CRC
Rani et al. 2017 [131]; Ran et al. 2014 [132]; Sebe et al. 2016 [133]	n-3 PUFAs& 5-FU	In vivo	Mouse	↓ Tumor burden and DNA damage ↓ Mucosal deterioration, ↑ Apoptosis ↓ 5-FU-related toxicity (intestinal mucositis) ↑ 5-FU anti-cancer activity
Ebadi et al. 2017 [134]	PUFAs & irinotecan	In vivo	Rat	Modulation of adipose tissue mitochondrial function ↓ 5-FU-associated side effects
Cai et al. 2014 [135]	n-3 PUFAs	In vitro	Human	↑ Lipid peroxidation, Modulation of the inflammatory response ↑ Apoptosis ↓ Cytotoxicity by radiation therapy
Granci et al. 2013 [136]	Fish oil& & 5-FU, oxaliplatin and irinotecan	In vitro	Human	↑ Apoptosis ↓ Cytotoxic effects of 5-FU, oxaliplatin and irinotecan.
Volpato et al. 2018 [113]	n-3 PUFAs	In vitro /In vivo	Human/ Mouse	↑ Butyrate-producing gut bacteria ↓ Inflammation; ↑ Apoptosis ↓ Proliferation of colon cancer cells
Watson et al. 2018 [115]	n-3 PUFAs	In vivo	Human	↓ Gut microbiota dysbiosis ↓ Pathogenic gut bacteria ↑ SCFA-producing gut bacteria (*Bifidobacterium*, *Roseburia* and *Lactobacillus*).
Piazzi et al. 2014 [120]	Eicosapentaenoic Acid	In vivo	Mouse	↑ *Lactobacillus* species in the gut microbiota ↓ Size of CRC tumors ↓ Proliferation colon cancer cells ↑ Apoptosis colon cancer cells
Song et al. 2015 [122]	n-3 PUFAs	In vivo	Human	↓ Risk of microsatellite instability ↑ DNA repair systems mismatch pathways
Yang et al. 2015 [124]	n-3 PUFAs	In vivo	Human	Different PUFA composition between normal and cancerous tissues ↓ Inflammation in CRC tumorigenesis.
Song et al. 2017 [126]	Marine ω-3 PUFAs	In vivo	Human	↑ Intake of marine ω-3 after CRC diagnosis ↓ Risk of CRC-specific mortality.
Aglago et al. 2020 [130]	n-3 PUFAs	In vivo	Human	Regular intake of fish at recommended levels ↓ Risk of CRC
Golkhalkhali et al. 2018 [137]	n-3 PUFAs & probiotic supplement	In vivo	Human	↑ Tolerability of capecitabine/oxaliplatin chemotherapy ↑ Quality of life markers ↓ Chemotherapy-induced symptoms (diarrhea and fatigue)
***Bioactive polyphenols***				
Mileo et al. 2019 [139]	Polyphenols	In vivo/In vitro	Human/ Mouse	↑ Gut microbiota balance ↓ Proliferation of colon cancer cells ↑ Apoptosis of colon cancer cells
Miene et al. 2009 [143]	Polyphenols (Apple)	In vitro	Human	Polyphenols are metabolized by colonic microbiota ↓ DNA damage induced by oxidative stress in colonic adenoma cells
Gibellini et al. 2011 [150]	Quercetin	In vivo	Human	↓ Proliferation of colon cancer cells ↑ Apoptosis of colon cancer cells
Venancio et al. 2017 [151]	Polyphenols (Cocoplum)	In vitro	Human	Anti-inflammatory activity and pro-oxidant effects
Lee Y et al. 2014 [152]	Apigenin	In vitro	Human	↓ Cell cycle progression ↓ Autophagy ↑ Apoptosis
Xavier et al. 2011 [160]	Polyphenols	In vitro	Human	↑ Apoptosis (in combination with 5-FU)
Hakim et al. 2014 [161]	Gelam Honey and Ginger	In vitro	Human	↑ Anticancer activity of 5-FU
Montrose et al. 2015 [162]	Black Raspberry (Anthocyanins, simple phenols, ellagic acid and quercetin)	In vivo	Mouse	↓ Expression of proinflammatory cytokines (TNF-α and IL-1β) ↓ Plasma levels of COX-2 and prostaglandin E2 ↑ Chemopreventive effect
McFadden et al. 2015 [163]	Curcumin	In vivo	Mouse	↑ Microbial diversity ↓ Colonic tumor burden ↑ Chemopreventive effect
Shakibaei et al. 2014 [164]	Curcumin	In vitro	Human	↑ Chemosensitization to 5-FU treatment
Buhrmann et al. 215 [165]	Resveratrol	In vitro	Human	↑ Chemosensitization to 5-FU treatment
Wang et al. 2015 [166]	Resveratrol	In vitro	Human	↓ Drug resistance (down-regulation of multi-drug resistant protein 1), ↓ Activation of NF-κB signaling ↓ Transcriptional activity of the cAMP-sensitive element
Paul et al. 2010 [153]	Pterostilbene (Blueberries)	In vivo	Rat	↓ Colon tumorigenesis by regulating the Wnt/b-catenin-signaling pathway ↓ Inflammatory responses.
Cui et al. 2010 [154]	Resveratrol	In vivo	Mouse	↓ Colitis-driven colon cancer incidence
Rodríguez-Ramiro et al. 2013 [155]	Polyphenols (Cocoa)	In vivo	Rat	Anti-inflammatory effect on the colonic tissue Chemoprevention in the early stages
Simons et al. 2009 [157]	Flavonol, Flavone and Catechin	In vivo	Human	↓ Risk of CRC
Zamora-Ros et al. 2017 [158]	Flavonoid	In vivo	Human	No association between regular dietary intake of flavonoids and CRC risk
Sánchez et al. 2019 [148]; Cueva et al. 2020 [149]	Red wine Polyphenols	In vivo	Human	Modulation of the gut microbiota composition ↓ Growth of pathogenic bacterial species (*F. nucleatum* and *P. Gingivalis*) ↓ Adhesion to oral cells ↓ Risk of CRC
***Probiotics***				
Hatakka et al. 2008 [169]	*Lactobacillus rhamnosus LC705* and *Propionibacterium freudenreichii* ssp.	In vivo	Human	Fecal counts of *Lactobacilli* and *Propionibacteria* ↓ β-glucosidase activity ↑ CRC prevention
Vinderola et al. 2006 [171]	*Lactobacillus kefiranofaciens*	In vivo	Human	Regulation of the immune system, ↑ Phagocytosis of tumor cells in early stages.
Galdeano et al. 2007 [172]	*Lactobacillus casei*	In vivo	Mouse	Induction of innate immunity influencing the clonal expansion of IgA B-cell population, ↓ Risk of CRC.
Bozkurt et al. 2019 [173]	*Bifidobacterium animalis subsp. lactis*	In vivo	Human/ Mouse	↑ Production mycosporin-like amino acids Modulation of the immune system to regulate the proliferation and differentiation of intestinal epithelial cells, macrophages, lymphocytes and cytokine production
Liu et al. 2011 [174]	*Lactobacillus plantarum, Lactobacillus acidophilus & Bifidobacterium Longum*	In vivo	Human	↑ Integrity of the intestinal barrier, ↑ Gut microbiota balance ↓ Post-operative infection rate
Rafter et al. 2007 [175]	*Lactobacillus rhamnosus & Bifidobacterium lactis*	In vivo	Human	Modulation of the gut microbiota composition ↓ Intestinal permeability, ↓ Cancer biomarkers (cell proliferation).
Hibberd et al. 2017 [176]	*Bifidobacterium animalis subsp. lactis Bl-04 & Lactobacillus acidophilus NCFM*	In vivo	Human	↑ Abundance of butyrate-producing bacteria in tumor, mucosa and fecal samples
Liang et al. 2016 [177]	*Bifidobacterium*	In vivo	Human	↓ Gut microbiota dysbiosis in CRC patients
Wan et al. 2014 [178]	*Lactobacillus delbrueckii*	In vitro	Cell line SW620	↓ Proliferation of colon cancer cells ↑ Apoptosis of colon cancer cells (via caspase 3 pathway)
Konishi et al. 2016 [179]	*Lactobacillus casei strain ATCC 334*	In vivo	Human	↑ Production of ferrichrome ↑ Apoptosis of colon cancer cells (via JNK pathway) ↓ Progression of CRC
Chang et al. 2018 [182]	*Lactobacillus casei Variety rhamnosus &* 5-FU/oxaliplatin	In vivo	Mouse	↓ Intestinal mucositis derived from anticancer treatment.
Ding et al. 2018 [183]; Lee et al. 2016 [184]; Routy et al. 2018 [185]	*Bifidobacterium &* PD-1-based immunotherapy	In vivo	Human/ Mouse	↓ Tumor growth ↓ Side effects induced by PD-1-based immunotherapy
Osterlund et al. 2007 [186]	*Lactobacillus rhamnosus*	In vivo	Human	↓ Side effects (severe diarrhea, abdominal distress) induced by chemotherapy.

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
