# Peer review of "The Role of the Gut Microbiome in Colorectal Cancer Development and Therapy Response"

_cancers, 2020, doi:10.3390/cancers12061406_

Round 1

Reviewer 1 Report

Dear authors, 

Your manuscript comprehensively review the role of the gut microbiome in the development and treatment of colorectal cancer (CRC), focusing specifically on the role that bacterial metabolites play in the immune response and inflammation processes in relation to CRC development and progression. It also highlights possible dietary interventions that could influence intestinal microbiota and CRC risk.  

I would recommend to mention in your manuscript as well the potential role that other components of the gut microbiota (rather than bacteria) could have in the development, progression and treatment of CRC (i.e. fungi, viruses, etc.), and which potential interventions could be considered. 

Best regards, 

Author Response

Reviewer 1

Comments to the Author

Dear authors, 

Your manuscript comprehensively review the role of the gut microbiome in the development and treatment of colorectal cancer (CRC), focusing specifically on the role that bacterial metabolites play in the immune response and inflammation processes in relation to CRC development and progression. It also highlights possible dietary interventions that could influence intestinal microbiota and CRC risk.  

I would recommend to mention in your manuscript as well the potential role that other components of the gut microbiota (rather than bacteria) could have in the development, progression and treatment of CRC (i.e. fungi, viruses, etc.), and which potential interventions could be considered. 

Response:We would like to thank the reviewer for their useful comments and suggestions that undoubtedly have helped to improve our manuscript. As suggested, in this new version of the manuscript we have added a section describing the potential role of gut virome and mycobiome in the development, progression and treatment of CRC (Page 3-4).

Reviewer 2 Report

#1. line 90 -92 “Ahn et al. described a decrease in bacterial diversity in fecal samples of CRC patients, with an increase in Fusobacterium nucleatum and Porphyromonas and a decrease in Gram-positive fiber-fermenting Clostridia [23].” >> or “Ahn et al. described a decrease in bacterial diversity in fecal samples of CRC patients, with an increase in Fusobacterium nucleatum and Porphyromonas and a decrease in Gram-positive fiber-fermenting Clostridia [24].”? And where is reference-23 mentioned in the text?

#2. line 99 -101 “In this regard, early signs of dysbiosis in adenoma and an increased abundance of F. nucleatum were associated to a higher expression of pro-inflammatory cytokines in colonic tissue from CRC patients [27-29].” >> Please double check. For example, the reviewer can’t find the term “nucleatum” in ref-29 [Cancer Epidemiol Biomarkers Prev 2017, 26(1), 85 - 94]

#3. line 133 – 137 “There are different pathogenic microbes associated to the promotion of CRC, including several Bacteroides species (B. vulgatus and B. stercoris), Bifidobacterium species (B. longun and B. angulatum), Eubacterium species (E. rectale 1 and 2, E. elignes 1 and 2, and E. cylindroides), Ruminococus species (R. torues, R. albus, and R. gnavus), Streptococo hansenii, Fusobacterium prausnitzi, and Peptoestreptococo productus 1 [40].” >> Please double check. For example, the reviewer can’t find the term “Bacteroides” or “vulgatus” in ref-40 [Cancer Prev Res (Phila) 2008, 1, 32-38]

#4. line 233 – 235 “In addition, the disruption of microbiota in MC38 colon carcinoma-grafted mice with a broad-spectrum antibiotics impaired tumor response to anti-CTLA4 immunotherapies [79].” >> Immunotherapy [intratumoral CpG-oligodeoxynucleotides, ODN] was used in ref-79 [Science 2013, 342, 967–970]. However, the reviewer is not sure whether ODN belongs to CTLA4 immunotherapy category or not.

#5. line 250 -253 “In addition, the increase bacterial fermentation of resistant starch to SCFAs (especially butyrate) has been shown to lower fecal pH in the colon, and this pH reduction can inhibit pathogenic organism proliferation and DNA damage induction, and enhance apoptosis and prevent proliferation of cancer cells [85-87].” >> The reviewer is not sure whether ref-86 [EMBO J 1990, 9(3), 849-855] is really related to this statement.

#6. line 271 – 273 “Finally, a recent meta-analysis has shown a CRC incidence reduction of 22% among pesco-vegetarian subjects compared with non-vegetarian individuals [99].” >> Please double check. Ref-99 [Trials 2019, 20(1), 688] was an ongoing clinical trial rather than a meta-analysis,

#7. line 278 – 281 “Pichard et al. showed that PUFAs have the potential to radio-sensitise HT29 colon cancer cells, possible due to an increase in lipid peroxidation products within the cells [104]. The same authors also reported an increase in apoptosis in colon cancer cells when combining 5-FU, oxaliplatin, and irinotecan with a fish oil emulsion with PUFAs [105]” >> Please note the 1st author of ref-104 [Clin Nutr 694 2014, 33(1), 164–170] & ref-105 [Br J Nutr 2013, 109(7), 1188] were Cai F & Granci V respectively, not Pichard, although Pichard C was the last author of both studies.

#8. line 363 – 365 “Previous studies have demonstrated that intestinal bacteria can repair injuries and reduce the incidence and severity of diarrea and bowel movements induced by the radiation therapy [136] (Fig.2).” >> Please note radiotherapy was not mentioned in figure -2.

#9. line 408 “Supplementary Materials: The following are available online at www.mdpi.com/xxx/s1,” >> The hyperlink www.mdpi.com/xxx/s1 doesn’t work. This file is also not available in  https://susy.mdpi.com/user/review/review/12052758/Ttq71DcV provided by the invitation email.

Author Response

Reviewer 2.

1. line 90 -92 “Ahn et al. described a decrease in bacterial diversity in fecal samples of CRC patients, with an increase in Fusobacterium nucleatum and Porphyromonas and a decrease in Gram-positive fiber-fermenting Clostridia [23].” >> or “Ahn et al. described a decrease in bacterial diversity in fecal samples of CRC patients, with an increase in Fusobacterium nucleatum and Porphyromonas and a decrease in Gram-positive fiber-fermenting Clostridia [24].”? And where is reference-23 mentioned in the text?

Response:We would like to thank the reviewer to identify the missing reference within the text. We have now included it after the mentioned sentence in the new version of the manuscript.

2. line 99 -101 “In this regard, early signs of dysbiosis in adenoma and an increased abundance of F. nucleatum were associated to a higher expression of pro-inflammatory cytokines in colonic tissue from CRC patients [27-29].” >> Please double check. For example, the reviewer can’t find the term “nucleatum” in ref-29 [Cancer Epidemiol Biomarkers Prev 2017, 26(1), 85 - 94]

Response:We would like to thank the reviewer for pointing out that this reference was not correctly cited here. Accordingly, we have now changed reference 29 by the following one: Yu T.; Guo F.; Yu Y.; Sun T.; Ma D.; Han J.; Qian Y.; Kryczek I.; Sun D.; Nagarsheth N.; Chen Y.; Chen H.; Hong J.; Zou W.; Fang J. Y. Fusobacterium nucleatum promotes chemoresistance to colorectal cancer by modulating autophagy. Cell 2017, 170, 548-563.e16. doi: 10.1016/j.cell.2017.07.008.

3. line 133 – 137 “There are different pathogenic microbes associated to the promotion of CRC, including several Bacteroides species (B. vulgatus and B. stercoris), Bifidobacterium species (B. longun and B. angulatum), Eubacterium species (E. rectale 1 and 2, E. elignes 1 and 2, and E. cylindroides), Ruminococus species (R. torues, R. albus, and R. gnavus), Streptococo hansenii, Fusobacterium prausnitzi, and Peptoestreptococo productus 1 [40].” >> Please double check. For example, the reviewer can’t find the term “Bacteroides” or “vulgatus” in ref-40 [Cancer Prev Res (Phila) 2008, 1, 32-38]

Response:Again, sorry for the mistake. The reference 40 (now reference 48) has been changed by the following one: Moore, W.E.; Moore, L.H. Intestinal floras of populations that have a high risk of CRC cancer. Appl. Environ. Microbiol. 1995, 61, 3202–3207

4.line 233 – 235 “In addition, the disruption of microbiota in MC38 colon carcinoma-grafted mice with a broad-spectrum antibiotics impaired tumor response to anti-CTLA4 immunotherapies [79].” >> Immunotherapy [intratumoral CpG-oligodeoxynucleotides, ODN] was used in ref-79 [Science 2013, 342, 967–970]. However, the reviewer is not sure whether ODN belongs to CTLA4 immunotherapy category or not.

Response:Thanks for your remark. The reference 79 (now reference 93) has been changed by the following one: Vetizou M. et al. Anticancer immunotherapy by CTLA-4 blockade relies on the gut microbiota. Science 2015, 350, 1079-1084. doi: 10.1126/science.aad1329.

5. line 250 -253 “In addition, the increase bacterial fermentation of resistant starch to SCFAs (especially butyrate) has been shown to lower fecal pH in the colon, and this pH reduction can inhibit pathogenic organism proliferation and DNA damage induction, and enhance apoptosis and prevent proliferation of cancer cells [85-87].” >> The reviewer is not sure whether ref-86 [EMBO J 1990, 9(3), 849-855] is really related to this statement.

Response:As indicated by the reviewer we have changed reference 86 (now reference 111) by the following one:Bergman EN. Energy contributions of volatile fatty acids from the gastrointestinal tract in various species. Physiol Rev. 1990 Apr; 70(2):567-90.

6. line 271 – 273 “Finally, a recent meta-analysis has shown a CRC incidence reduction of 22% among pesco-vegetarian subjects compared with non-vegetarian individuals [99].” >> Please double check. Ref-99 [Trials 2019, 20(1), 688] was an ongoing clinical trial rather than a meta-analysis,

Response:Thank you for the comment. As indicated by the reviewer we have changed “meta-data” by “clinical trial” in the revised version of the manuscript (line 396).   

7. line 278 – 281 “Pichard et al. showed that PUFAs have the potential to radio-sensitise HT29 colon cancer cells, possible due to an increase in lipid peroxidation products within the cells [104]. The same authors also reported an increase in apoptosis in colon cancer cells when combining 5-FU, oxaliplatin, and irinotecan with a fish oil emulsion with PUFAs [105]” >> Please note the 1stauthor of ref-104 [Clin Nutr 694 2014, 33(1), 164–170] & ref-105 [Br J Nutr 2013, 109(7), 1188] were Cai F & Granci V respectively, not Pichard, although Pichard C was the last author of both studies.

Response:Thank you for the comment. We have now corrected the names of the first authors in both references (now references 137 and 138, respectively).

8. line 363 – 365 “Previous studies have demonstrated that intestinal bacteria can repair injuries and reduce the incidence and severity of diarrea and bowel movements induced by the radiation therapy [136] (Fig.2).” >> Please note radiotherapy was not mentioned in figure -2.

Response:We would like to thank the reviewer for his/her remark. We added figure 2 to its correct position in the manuscript (line 478).

9. line 408 “Supplementary Materials: The following are available online at www.mdpi.com/xxx/s1,” >> The hyperlink www.mdpi.com/xxx/s1 doesn’t work. This file is also not available in https://susy.mdpi.com/user/review/review/12052758/Ttq71DcV provided by the invitation email.

Response: We are sorry for the misunderstanding. There is no supplementary material in the review. This paragraph has been removed from the revised version of the manuscript.

Reviewer 3 Report

The review written by Sanchez-Alcoholado and colleagues covers research on the gut microbiome in CRC. This is an emerging field in the context of CRC and thus relevant to the research community.

The review reads well and tries to cover a very wide spectrum of subtopics. Nevertheless, the reviewer would like to suggest a few changes.

  • As the authors try to cover a very broad research field, they mention a lot of different subtopics and end up only scratching the surface of these numerous subtopics without going into the detail of the most interesting ones. For example, the first pages (introduction and paragraph 2) have been addressed in multiple reviews and could be shortened. On the other hand, paragraph 3 (partially) but most importantly paragraph 5 are not always covered in reviews and could include a more extensive literature review. This is a special concern as the title presumes that the majority of the review will be about the dietary mediators. But at the end it only broadly describes it without any extensive discussion. Here the reviewer would suggest a table mentioning the most important related studies (especially in vivo studies).
  •  
  • Additionally and along what was mentioned above (dietary mediators), in its present format, the authors should think about changing the title to avoid any false impression on the review content.

  • Some new meta-analysis studies of antibiotics treatment and its impact on CRC are missing.

  • Additionally, according to GLOBOCAN 2018 data, CRC is the third most deadly cancer in the world. Please correct the sentence "Colorectal cancer (CRC) is the third most common cancer worldwide and the leading cause of cancer-related deaths." It is not the leading cause of cancer-related deaths.

  • The reviewer would also suggest to change the following sentence and rewrite it more cautiously. “These changes can produce the enrichment of pro- inflammatory opportunistic pathogens and the decrease of butyrate-producing bacteria, leading toan imbalance in intestinal homeostasis that can lead to tumor formation." The latter part should be reformulated as causality (i.e. dysbiosis leads to) has not been proven. 

Author Response

Reviewer 3

The review written by Sanchez-Alcoholado and colleagues covers research on the gut microbiome in CRC. This is an emerging field in the context of CRC and thus relevant to the research community. The review reads well and tries to cover a very wide spectrum of subtopics. Nevertheless, the reviewer would like to suggest a few changes.

1. As the authors try to cover a very broad research field, they mention a lot of different subtopics and end up only scratching the surface of these numerous subtopics without going into the detail of the most interesting ones. For example, the first pages (introduction and paragraph 2) have been addressed in multiple reviews and could be shortened. On the other hand, paragraph 3 (partially) but most importantly paragraph 5 are not always covered in reviews and could include a more extensive literature review. This is a special concern as the title presumes that the majority of the review will be about the dietary mediators. But at the end it only broadly describes it without any extensive discussion. Here the reviewer would suggest a table mentioning the most important related studies (especially in vivo studies).

Response:We thank the reviewer for their useful comments and suggestions. If the reviewer agrees we would prefer not to shorten the introduction section because we think that the information is important not only for the revision format but also to understand the rest of sections included in the manuscript. On the other hand, in this new version we have inserted further information about “dietary mediators” (paragraph 5) based on a more extensive literature. Finally, we have added a table mentioning the most important related in vivo/in vitro studies as suggested by the reviewer.

2. Additionally and along what was mentioned above (dietary mediators), in its present format, the authors should think about changing the title to avoid any false impression on the review content.

Response:Thank you for the remark. As suggested, we have modified the title as follows: “The Role of the Gut Microbiome in Colorectal Cancer Development and Therapy Response”

3. Some new meta-analysis studies of antibiotics treatment and its impact on CRC are missing.

Response:We thank the reviewer for his/her suggestion. We have now included some new meta-analysis studies of antibiotics treatment and its impact on CRC (Page 7).

4. Additionally, according to GLOBOCAN 2018 data, CRC is the third most deadly cancer in the world. Please correct the sentence "Colorectal cancer (CRC) is the third most common cancer worldwide and the leading cause of cancer-related deaths." It is not the leading cause of cancer-related deaths.

Response:According to the reviewer’s comment we have modified the sentence in the revised version of the manuscript (line 71).

5. The reviewer would also suggest to change the following sentence and rewrite it more cautiously. “These changes can produce the enrichment of pro- inflammatory opportunistic pathogens and the decrease of butyrate-producing bacteria, leading to an imbalance in intestinal homeostasis that can lead to tumor formation." The latter part should be reformulated as causality (i.e. dysbiosis leads to) has not been proven. 

Response:We have now reformulated the sentence as indicated by the reviewer: “These changes might produce enrichment in pro-inflammatory opportunistic pathogens and a decrease in butyrate-producing bacteria, which may lead to an imbalance in intestinal homeostasis (dysbiosis) that could ultimately lead to tumor formation” (lines 87-90).

Round 2

Reviewer 2 Report

I have no additional comments.